# Splicing Modulators Are Involved in Human Polyglutamine Diversification via Protein Complexes Shuttling between Nucleus and Cytoplasm

**DOI:** 10.3390/ijms24119622

**Published:** 2023-06-01

**Authors:** Makoto K. Shimada

**Affiliations:** Center for Medical Science, Fujita Health University, Toyoake 470-1192, Japan; mshimada@fujita-hu.ac.jp

**Keywords:** polyglutamine (polyQ), protein complex, intrinsically disordered region, hub protein

## Abstract

Length polymorphisms of polyglutamine (polyQs) in triplet-repeat-disease-causing genes have diversified during primate evolution despite them conferring a risk of human-specific diseases. To explain the evolutionary process of this diversification, there is a need to focus on mechanisms by which rapid evolutionary changes can occur, such as alternative splicing. Proteins that can bind polyQs are known to act as splicing factors and may provide clues about the rapid evolutionary process. PolyQs are also characterized by the formation of intrinsically disordered (ID) regions, so I hypothesized that polyQs are involved in the transportation of various molecules between the nucleus and cytoplasm to regulate mechanisms characteristic of humans such as neural development. To determine target molecules for empirical research to understand the evolutionary change, I explored protein–protein interactions (PPIs) involving the relevant proteins. This study identified pathways related to polyQ binding as hub proteins scattered across various regulatory systems, including regulation via PQBP1, VCP, or CREBBP. Nine ID hub proteins with both nuclear and cytoplasmic localization were found. Functional annotations suggested that ID proteins containing polyQs are involved in regulating transcription and ubiquitination by flexibly changing PPI formation. These findings explain the relationships among splicing complex, polyQ length variations, and modifications in neural development.

## 1. Introduction

Polyglutamine (polyQ) functions as a binding domain that plays a crucial role in protein complex formation. Our previous research using a human gene database (H-InvDB) indicated that polyQ regions are observed at significantly higher rates in genes that are functionally annotated as being involved in the modification of neural development [1]. Nine of these polyQs are known to be causative of neurodegenerative diseases, in which longer repeat alleles confer a higher risk of disease onset. Considering the diversified length polymorphisms at these disease-causing polyQ repeats in humans compared with those in apes [2], my colleagues and I hypothesized that this diversification may be advantageous in humans based on the finding that these genes are strongly associated with neural development and what we know about human evolution [1]. This includes the finding that variation in polyQ length may have led to the diversification of personalities (individuality) through diversification of the neural development process, which may in turn be advantageous for building complex structured human populations with sophisticated cultures. However, the molecular basis linking these two variations is unknown, and there is also no information on the molecular pathways involved to perform empirical research. To confirm the above hypothesis, it is necessary to clarify the pathways by which human personalities diversify through the regulation of neural development. This study aims to select target molecular pathways by which polyQ length polymorphisms affect phenotypes of neural development.

This study focuses on information of protein–protein interactions (PPIs), with the expectation that this will provide clues for detecting the pathways or protein complexes involved in mechanisms regulating neural development that may be affected by polyQ length variation. Information regarding the subcellular localization of the protein complexes is also important for identifying target pathways to be studied. This is because the information that a molecule localizes in both the nucleus and cytoplasm suggests a function of the transportation between them and post-transcriptional modification [3]. Recent progress in our understanding of the complex formation and intrinsically disordered (ID) proteins is changing views about the 3D conformation of protein complexes [4,5,6]. The flexibility and promiscuity of ID proteins in binding multiple partners have led to the realization that ID proteins can be hub proteins. Considering that polyQ regions form ID regions and bind multiple molecules as hub proteins, proteins with polyQ and polyQ-binding proteins are expected to be involved in post-transcriptional modification or signaling between the nucleus and cytoplasm. This may lead to the regulation of neural development.

Because all polyQ repeat diseases are known to be specific to humans, emerging risk alleles containing long polyQ repeats may be subjected to human-specific selective pressure. Considering base-substitutional genomic changes generally evolve slowly and the genomic difference between humans and chimpanzees is about 1%, such human-specific selective pressure should be detected on small genomic changes that produce drastic changes in phenotypes. Alternative splicing is a source of phenotypic diversity by generating multiple transcripts from a single gene. If frequent changes in the repeat length of polyQ affect alternative splicing in genes involved in regulating the process of a human-specific feature, human-specific neuronal changes are likely to emerge in a short period in human evolution.

Splicing modulators act in protein complexes known as splicing complexes [7]. It is anticipated that splicing modulators within splicing complexes (spliceosomes) are influenced by polyQ-containing proteins. Indeed, deregulated splicing is considered to be a source of RNA toxicity in one polyQ disease, Huntington’s disease (HD) [8]. A polyQ-containing protein, huntingtin (HTT), is known to be causative of HD. HTT is a hub protein that binds various partner proteins and plays multiple roles via various post-translational modifications and the formation of various complexes. These roles include activities in vascular transport, cell division, and ciliogenesis, in which information on the subcellular localization is key for estimating the function of the HTT protein [9]. The information that HTT adopts both nuclear and cytoplasmic localizations is particularly important for specifying its functions regarding the regulation of gene expression.

The aim of this study is to determine the function of polyQ regions in protein complexes, with a special focus on splicing modulators that bind to polyQs. PQBP1 is the first polyQ-binding protein to be surveyed. PQBP1 is known to interact with polyQs, with a higher affinity to longer stretches of disease-causing repeats in HTT, androgen receptor (AR), and ataxin 1 (ATXN1) [10,11]. PQBP1 is also a spliceosomal protein that forms part of the spliceosomal complex [12,13]. Besides PQBP1, TERA/VCP/P97 (VCP) is also a well-documented polyQ-binding protein. I selected these two proteins as representative polyQ-binding proteins based on the relative abundance of available information about them. This study collected information on PPIs and pathways regarding these polyQ-binding proteins.

## 2. Results

### 2.1. Search for PPI-Related Information on Complex Formation and Biological Functions Focusing on PolyQ-Binding Ability

A search in the database HIPPIE [14] for the PPIs of the two representative polyQ-binding proteins (binders), namely, PQBP1 and VCP, identified 55 proteins for PQBP1 and 620 proteins for VCP in total. Among these proteins, three (ATXN1, AR, and HTT) and five (ATXN3, HTT, AR, ATXN1, and ATXN7) disease-causing polyQ-containing proteins (carriers) were included in PPIs for PQBP1 and VCP, respectively. Then, a search for PPIs of the carriers was conducted and proteins that interact with both the binders and the carriers (co-interactors) were extracted (Table 1a and Appendix A).

Meanwhile, co-interactors acting as a bridge between the binders and the carriers obtained by multiple searches of STRING [15] were different from the results using HIPPIE (Table 1b and Appendix A). Although three to five co-interactors linking VCP to HTT, AR, and ATXN3 were also included in the corresponding results of HIPPIE, there were no co-interactors in common for PQBP1 (Appendix A). The difference between PQBP1 and VCP in terms of their interaction patterns was found in both PPI database search results (Figure 1 and Appendix A).

The features of these co-interactors were obtained by gene enrichment analysis using DAVID [16] (Table 2 and Appendix A). Co-interactors that interact with PQBP1 are associated with the function “Regulation of transcription”, while those that interact with VCP are associated with “Ubiquitin-proteasome for ER stress”. Moreover, both groups have in common the molecular function “Protein binding” and the cellular components “Nucleoplasm” and “Cytoplasm”.

### 2.2. Search for PPI Information Focusing on Nucleus- and Cytoplasm-Localized Intrinsically Disordered (ID) Hub Proteins

Nine polyQ-containing proteins were found in the list of the nucleus- and cytoplasm-localized ID hub proteins [polyQ-containing ID hub proteins localized in both nucleus and cytoplasm (NC-ID-hub)] (Table 3 and Appendix A), which is significantly more than would be expected from a random selection (Fisher’s test, *p* < 0.05). This indicates that the proteins with a polyQ region may be particularly common within complexes, which would be due to the expected structural advantage of ID regions and the localization in the nucleus and the cytoplasm.

The PPIs found in this study suggest that these nine polyQ-containing proteins play roles in multiple signaling pathways (Table 3). HTT is known as a scaffold for forming protein complexes [9]. Our finding that the polyQ-containing proteins are observed significantly more frequently among ID hub proteins localized in both the nucleus and the cytoplasm indicates that polyQ-containing proteins other than HTT also play significant roles in signal transduction between the nucleus and the cytoplasm utilizing the feature of intrinsic disorder in the polyQ region.

For these nine proteins with a polyQ region, eight PPIs were found (Figure 2A). They were found to be involved in signaling pathways such as Notch signaling, ubiquitination, and histone acetylation. Among these eight PPIs, six [(1–4), (7), and (8)] were between proteins in different repeat-length polymorphism (RLP) categories, namely, polymorphic/monomorphic, while the other two PPIs [(5) and (6)] were between proteins in the same RLP category (i.e., within the polymorphic or monomorphic category) (Figure 2A).

Notably, for one of the obtained PPIs, between ATXN1 and USP7, it is known that the strength of the interaction is dependent on the polyQ length. The C-terminal region of ATXN1 is also known to be essential for the interaction, although the polyQ region is located at the N-terminal [17]. Consequently, the signaling pathways involving these two proteins were investigated in the literature and a pathway involved in the regulation of transcription by deubiquitination was identified (Figure 2B).

To understand the relationship between ID regions and function, the nine proteins were used as queries in the Database of Disordered Protein Predictions (D^2^P^2^), which integrates multiple disordered predictions and structural domains. This showed three patterns of positional relationship between polyQs and “Molecular Recognition Features” (MoRFs) as predicted by ANCHOR for binding regions in ID regions [18]. First, polyQs were held by MoRFs in five proteins (ATXN1, AR, NCOA3, EP300, and CREBBP). This positioning is likely to facilitate the latching of the encountered proteins into complexes. As prominent examples, three of them (NCOA3, EP300, and CREBBP) are known to resemble each other in the structure of their domains and their functions as nuclear receptor coactivators, in which polyQs are located close to nuclear receptor coactivator interlocking domains and facilitate the interaction between transcription factors and transcriptional machinery [19,20,21]. Second, a polyQ region partially overlapped with a MoRF in HTT. Third, polyQs completely overlapped with MoRFs in USP7 and RNABP9 (Appendix A). Because MoRFs are binding regions in ID regions and these nine polyQs were located within predicted ID regions, these polyQs are located in regions that are important for the functions of hub proteins that regulate multiple reactions by forming multiple complexes.

### 2.3. Review of Signaling Pathways That Explain Effect of Huntingtin (HTT) PolyQ Length on Neurite Length

Mehta et al. (2018) [22] found a negative correlation between polyQ repeat length of HTT and neurite length after differentiation (130 days) of induced pluripotent stem (iPS) cells into cortical neurons. The polyQ repeat length consisted of six groups, three in the “normal” range (18, 21, and 33 repeats) and three “abnormally hyper-expanded” groups (77, 109, and 180 repeats). Based on this negative correlation, we summarized the relationship between the signals regulating neurite length and the polyQ length of HTT during neuronal development, based on information in the literature (Figure 3). Notably, through the literature search, one of the polyQ-binding proteins, VCP, was extracted as playing a role in the pathway from HTT to the activation of Rac, followed by the formation of lamellipodia and promotion of neurite outgrowth.

## 3. Discussion

This study illustrated how ID proteins could be involved in the relationship between polyQ length polymorphism and the human evolution of individuality. Specifically, polyQs form sequences that are rich in polar amino acids, which leads to the formation of ID regions [23]. Although ID regions flexibly and transiently change their conformation, MoRFs in ID regions play regulatory roles by forming determinate structures in complex assembly at specific times and locations [24,25]. This study shows that some polyQ regions play crucial roles in the formation of complexes carrying various factors and signals by effectively utilizing the characteristics of ID regions in the 3D structure.

### 3.1. PolyQ Repeat Length Variation

In accordance with the hypothesis that polyQ length polymorphisms brought about personality differences, it is known that the affinity of interaction changes according to the polyQ repeat length for HTT, ATXN1, and AR. The N-terminal of HTT contains a polyQ region that interacts with the nucleoprotein TPR, one of the nuclear pore complexes, for its export from the nucleus to the cytoplasm. PolyQ length expansion has been reported to decrease this interaction [26]. Moreover, an association between the polyQ repeat length of HTT and the size of gray matter in the human brain has been reported, along with a discussion of the possibility that polyQ length variation of HTT alters the affinity of interactions with transcription factors through conformational difference [27]. In ATXN1, the affinity of binding to RNA decreases as the polyQ length increases, which was proposed to affect function in RNA metabolism [28]. Like changes in affinity in a manner dependent on PQBP1 length [10], AR polyQ length polymorphism is widely assumed to affect body composition, bone metabolism, psychiatric status, male sexual function and fertility, cardiovascular risk, the risk of prostate and testicular cancer, the risk of systemic lupus erythematosus in females, as well as the risk of the neurodegenerative disease spinal and bulbar muscular atrophy (SBMA, also known as Kennedy disease) [29,30,31].

### 3.2. Difference between the Two PolyQ-Binding Proteins

PQBP1 has been shown to be related to transcriptional regulation in brain development (Figure 1, Table 2). This is interesting given that PQBP1 is known to bind splicing factors [13,32] and induce alternative splicing patterns, which in turn alters neurite outgrowth [33,34]. Meanwhile, VCP, which is closely related to pathogenesis, has been shown to be involved in quality control in the endoplasmic reticulum via the ubiquitin-proteasome system (Table 2).

Wang et al. (2013) [33] presented that a binder molecule, PQBP1, regulates neurite outgrowth length, but the process by which PQBP1 binds the polyQ of HTT was not included in the study design. Meanwhile, Mehta et al. (2018) [22] demonstrated a negative correlation between the polyQ length variation of HTT and neurite length variation, but no polyQ binder molecule was known (Figure 4). Considering that alternative splicing is a molecular mechanism to generate phenotypic diversity and PQBP1 regulates alternative splicing, it may be natural to consider that both results should be due to the same pathway. However, this review of information regarding HTT and neurite outgrowth suggests that signal processing via VCP, not PQBP1, as a binding protein to polyQ of HTT generates a negative correlation (Figure 3). This shows that, because PQBP1 is a splicing factor, it does not always follow that PQBP1 regulates phenotypic variation such as neurite length.

### 3.3. Suitable Protein Complex

The objective of this study is to identify an appropriate protein complex to be experimentally used for the further empirical study that clarifies the relationship between polyQ length and a phenotype that has been targeted by natural selection during human evolution. Generally, pathways that are active within limited tissues and timepoints are more suitable as target pathways for experimental operation than ones that are ubiquitously active because effects of operations such as knockdown and rescue experiments can be easily traced in limitedly active pathways. Proteins involved in PPIs with VCP included more hub proteins involved in ubiquitous pathways than those interacting with PQBP1. Accordingly, pathways associated with proteins undergoing PPIs with PQBP1 may be more suitable for empirical study.

### 3.4. Traceability of PPI Information

The traceability of algorithms and the data source used in PPI searches are important for applying the obtained results, even in the artificial intelligence era. Among the co-interactors found through PPI searches for both carriers and binders (Table 1), only CREBBP was also found among the ID hub proteins (Table 3). This scarcity indicates that polyQs may be involved in so many interactions that the proteins identified as PPI partners vary depending on how the search is conducted. The differences in search results between the two PPI databases (HIPPIE and STRING) are due to the data and algorithm used. Specifically, HIPPIE performs collection and scoring mainly based on experimental evidence and provides a context-specific network [14]. However, this is based on the tendency for disease-causing proteins to form tissue-specific PPIs more in the tissues that they affect than in other tissues [35]. Considering that this study focuses on the evolutionary mechanism leading to diverse polymorphisms, filtering by function as implemented in HIPPIE is useful to avoid the bias associated with the particular abundance of disease-related information that has been accumulated. [This abundance is highlighted by all PPIs of AR being filtered out when brain-specific PPIs were selected (Appendix A).] Upon narrowing down the HIPPIE PPI networks to those expressed in the brain, the number of AR interactors decreased from 131 to zero (Appendix A). Because AR plays multiple roles including in hormonal regulation, this contrast with other disease-causing polyQ proteins may indicate that behavioral variations relating to AR occur via a different molecular mechanism from those of other polyQ proteins. Meanwhile, STRING collects all types of publicly available PPI data with the aim of achieving wide coverage by incorporating prediction and existing large datasets. These differences between the two PPI databases may suggest that there are differences between them in the definition of PPIs used, which includes the types of interactions, such as types of chemical bonds and association via liquid–liquid phase separation.

## 4. Materials and Methods

To obtain clues to help understand how the polyQ length polymorphism brought about personality differences or individuality in humans, this study was conducted using three approaches: (1) a search for information on protein–protein interactions (PPIs) to identify complex formation and biological functions associated with polyQ-binding ability; (2) a search for information on PPIs focusing on nucleus- and cytoplasm-localized ID hub proteins; and (3) a review of signaling pathways that explain the effect of HTT polyQ length on neurite length.

(1)A search for information on PPIs associated with the complex formation and biological functions focusing on polyQ-binding ability:

First, a search of the PPIs of PQBP1 and VCP as two representative polyQ-binding proteins (binders) that bind to polyQ was performed in the PPI database HIPPIE (v2.1) [14,36]. A search of the PPIs of polyQ-disease-causative proteins (carriers) was also performed in the same way. To understand the robustness of the PPI searches, a search of co-interactors that interact with both binders and carriers was also performed, inputting a binder and a carrier as a query in the multiple search tool STRING (v11.5) [15]. To obtain information on the biological pathway and function shared among the carrier, the binder, and the co-interactors, Gene Ontology information of the co-interactors obtained in the search of the HIPPIE database was extracted using DAVID (v6.8) [37].

(2)A search for information on PPIs focusing on nucleus- and cytoplasm-localized ID hub proteins:

To identify protein complexes and their biological pathways with which polyQ-containing proteins are involved, we focused on the feature that polyQ regions are ID regions that are known to be advantageous for binding to various types of 3D structures of proteins. This was based on the assumption that this feature is one of the molecular bases explaining why polymorphic polyQ proteins are associated with the regulation of neurodevelopment, as shown in our previous study [1]. Considering that protein complexes acting in the regulation of neurodevelopment need to regulate expression by transducing signals between the nucleus and cytoplasm, a search of polyQ-containing ID hub proteins that localize in both nucleus and cytoplasm (NC-ID-hub) was performed as follows.

First, two gene lists were compared. One featured all 198 polyQ-containing proteins listed in our previous study [1]. The other was a list of the 210 hub proteins listed as being localized in both the nucleus and the cytoplasm by Ota et al. (2016) [3]. The overlapping proteins were defined as NC-ID-hub polyQ-containing proteins. The number of overlaps was statistically evaluated by comparison with the distribution of randomly occurring overlaps that were obtained from 10,000 simulations of overlap between the 198 polyQ-containing genes and 210 randomly selected proteins from among all human protein-coding genes. The extracted proteins were summarized using annotations in life science databases, UniProt (release 2017_12) [38], H-InvDB (v9.0) [39], and Human Protein Reference Database (HPRD) (v9) [40]. A search of the interactions among the obtained proteins was performed using the tab listing “Protein Interactions” in the HPRD.

The Database of Disordered Protein Predictions (D^2^P^2^) (release 2013_01) was used to determine the relationship between ID and structured domains in each protein. Protein identifiers of UniProt (i.e., UniProt accessions such as P42858) [38] were used as queries for the D^2^P^2^ search.

(3)Review of signaling pathways that explain the effect of HTT polyQ length on neurite length:

Information was collected on the following: cellular process involving HTT, neurite outgrowth regulation and intracellular signal pathway in the developmental process of neural cells. Based on this information, molecular mechanisms or signal pathways that are possibly involved in neurite outgrowth in humans were selected. Within these signal pathways, one in which proteins interacting with HTT play roles was selected and cascades of the reactions that connect HTT and the regulation of neurite outgrowth were constructed.

## 5. Conclusions

This study provides clues to confirm the molecular basis linking the variation in polyQ length and the variation associated with human individuality, namely, the generation of variation in neurite outgrowth, which is regulated by various protein complexes including spliceosome proteins. Considering that HTT is a hub protein that plays various roles, the polyQ region of HTT is likely to have been subjected to multiple selective pressures depending on the cell type, tissue, and developmental stage. A possible pathway by which HTT polyQ length affects neurite length is a VCP-associated signal cascade (Figure 3). Meanwhile, in the analysis focusing on NC-ID-hub proteins, PQBP1 was shown to be involved in a pathway in which CREBBP acts as a co-interactor with HTT (Figure 2A, Appendix A). Based on PQBP1 being a splicing factor, this indicates that the two abovementioned variations can be splicing-mediated. Other polyQ-containing NC-ID-hub proteins such as ATXN1 and AR also interact with CREBBP and PQBP1 (Appendix A), which may suggest various similar PPIs depending on the regulatory context. There is a need for further studies aimed at detecting the changes in selective pressure depending on differences in the context of polyQ-coding regions and their neighboring genomic regions.

## Figures and Tables

**Figure 1 ijms-24-09622-f001:**
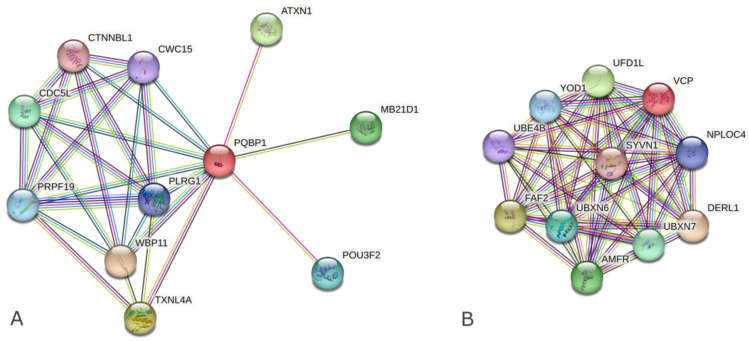
PPI networks among polyQ-binding proteins using STRING. (**A**) PQBP1, (**B**) VCP. Default settings of the position of nodes and colors of edges are left unchanged, which means that nodes have been placed at minimizing the ‘energy’ of the system of STRING and colors of edges indicates types of evidence as follows: Red line—indicates the presence of fusion evidence, Green line—neighborhood evidence, Blue line—cooccurrence evidence, Purple line—experimental evidence, Yellow line—text-mining evidence, Light blue line—database evidence, Black line—coexpression evidence.

**Figure 2 ijms-24-09622-f002:**
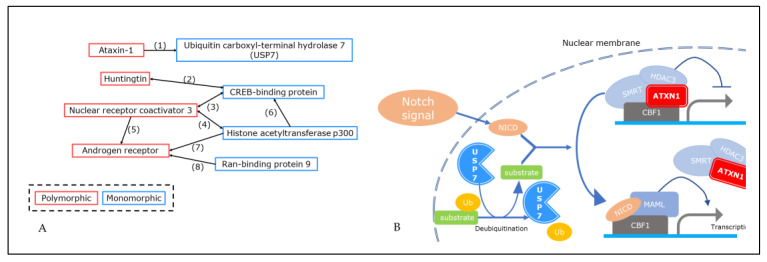
(**A**) Interactions found in polyQ-containing ID hub proteins located in both cytoplasm and nucleus using protein interaction annotation of HPRD. Direction of arrow indicates the protein interaction annotation used (i.e., “A → B” depicts that the HPRD annotation for protein interactors of protein A includes protein B; double-headed arrow depicts PPIs found in both proteins). Red and blue boxes indicate polymorphic and monomorphic in polyQ repeat length, respectively. (**B**) Ataxin-1 and USP7 participate in the same signaling pathway.

**Figure 3 ijms-24-09622-f003:**
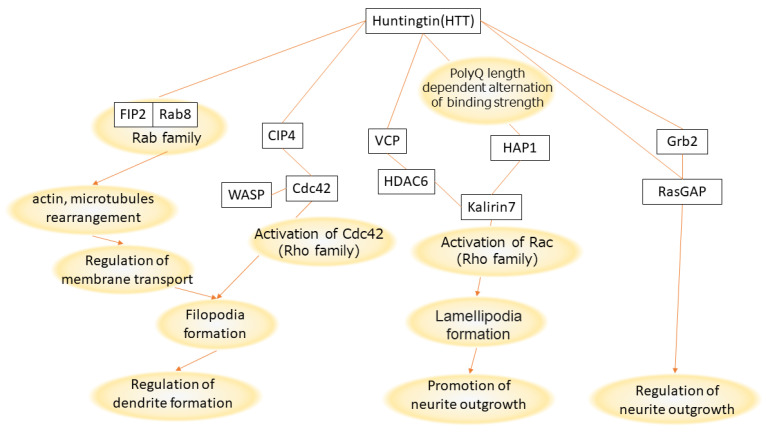
Possible pathways by which HTT polyQ length variation generates variation of neurite outgrowth.

**Figure 4 ijms-24-09622-f004:**
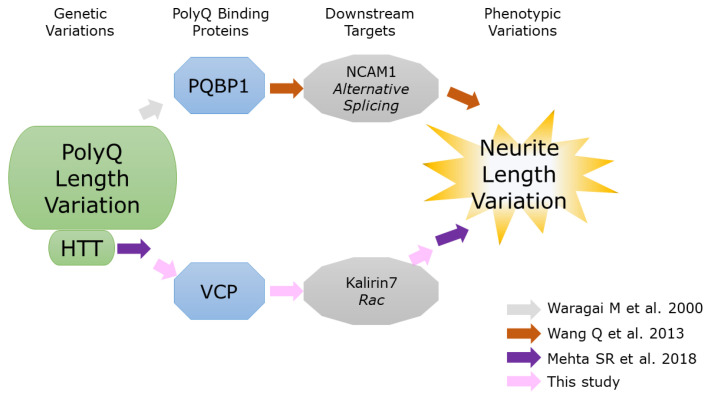
Relationship among pathways demonstrated by each study regarding HTT polyQ length variation and neurite length variation [12,22,33].

**Table 1 ijms-24-09622-t001:** (a) Number of proteins that interact with both polyQ-binding proteins and polyQ-disease-causing proteins, determined using HIPPIE. (b) Number of proteins that interact with both polyQ-binding proteins and polyQ-disease-causing proteins, determined using STRING.

**(a)**										
		**PolyQ-Disease-Causing Proteins (Carriers)**								
		**HTT**	**AR**			**ATXN1**	**ATXN3**			**ATXN7**
Binding proteins (binders)	PQBP1	2	4			1	0			0
	VCP	52	44			25	34			11
**(b)**										
		**PolyQ-Disease-Causing Proteins (Carriers)**								
		**HTT**	**AR**	**ATN1**	**ATXN1**	**ATXN2**	**ATXN3**	**ATXN7**	**CACNA1A**	**TBP**
Binding proteins (binders)	PQBP1	2	3	3	9	8	0	2	1	4
	VCP	10	6	4	5	7	29	5	2	12

**Table 2 ijms-24-09622-t002:** Features of co-interactor proteins suggested by gene enrichment analysis.

		Features	
		Specific	Shared
Binding proteins (binders)	PQBP1	Regulation of transcription	Nucleoplasm (CC) Cytoplasm (CC) Protein binding (MF)
	VCP	Ubiquitin-proteasome for ER stress	

**Table 3 ijms-24-09622-t003:** (a) Nucleus- and cytoplasm-localized ID hub proteins that contain polymorphic polyQ in repeat length ^(1)^. (b) Nucleus- and cytoplasm-localized ID hub protein that contain monomorphic polyQ in repeat length ^(1)^.

**(a)**				
**Specific GO ^(2)^**	**Protein Name**	**# PPIs ^(3)^**	**Molecular Function**	**Biological Pathway**
Present	Ataxin-1 (ATXN1)	159	RNA binding	Regulation of nucleic acid metabolism
	Androgen receptor (AR)	150	Ligand-activated receptor	Signal transduction
	Huntingtin (HTT)	59	DNA binding	Regulation of nucleic acid metabolism
Absent	Nuclear receptor coactivator 3 (NCOA3)	47	Transcriptional regulation	Intracellular signaling
**(b)**				
**Specific GO ^(2)^**	**Protein Name**	**# PPIs ^(3)^**	**Molecular Function**	**Biological Pathway**
n/a	Histone acetyltransferase p300 (EP300)	209	Transcriptional regulation	Regulation of nucleic acid metabolism
	CREB-binding protein (CREBBP)	198	Transcriptional regulation	Regulation of nucleic acid metabolism
	Ubiquitin carboxyl-terminal hydrolase 7 (USP7)	40	Deubiquitinating enzyme	Protein metabolism
	Ran-binding protein 9 (RANBP9)	28	Enzyme binding	Protein-containing complex assembly
	Heterogeneous nuclear ribonucleoprotein D0 (HNRNPD)	19	mRNA binding	Regulation of nucleic acid metabolism

^(1)^ Presence or absence of repeat-length polymorphism (RLP). At least a single polymorphic polyQ region is present (polymorphic) or absent (monomorphic). ^(2)^ Presence or absence of polymorphic polyQ-specific GO annotation (i.e., neurodevelopmental regulation) in the gene as determined by data from Shimada et al. (2016) [1]. ^(3)^ Number of protein–protein interactions (PPIs) excluding duplications from the list of Ota et al. (2016) [3].

## Data Availability

Data are contained within the article or Appendix A. The data presented in this study are available in Shimada et al. (2016) [1] and Ota et al. (2016) [3].

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
