# Peer review of "Splicing Modulators Are Involved in Human Polyglutamine Diversification via Protein Complexes Shuttling between Nucleus and Cytoplasm"

_ijms, 2023, doi:10.3390/ijms24119622_

Round 1

Reviewer 1 Report

In this study, the authors sought to identify the roles of polyQ regions in protein complexes involved in numerous biological processes.

It is advised that the writers respond to the following criticisms.

1. Poorly written abstract. Instead of just listing them randomly, the authors should make a precise outline of their objectives and ambitions.

2. Writers should carefully examine their spelling. There are numerous spelling mistakes. For instance, on lines 8 and 118, respectively, correct the spellings of "polyglutamin" and "althernation."

3. The English utilized in this book is generally of very low quality. For instance, the language in lines 96–98 and 131–136 is unclear and confusing. The author should use someone who is proficient in English to edit their manuscript.

4. Why are PQBP1 and VCP preferred to other polyQ-binding proteins in the hunt for PPIs?

5. Why are there no interactions for AR in figure 1?

6. What do the single- and double-headed arrows in image 2a represent?

Reviewer 2 Report

The manuscript by Shimada et al. (Manuscript ID: ijms-2283380) titled "Splicing modulators involve in human polyglutamine diversification via protein complexes shuttling between nucleus and cytoplasm" aimed to determine the function of the polyQ region in protein complexes. Using databases, authors report that polyQs function as scaffolds of complex formation carrying various signals. The manuscript is providing some results on protein complexes and splicing modulators that bind to polyQs. I have some comments as follows:

-    Authors mentioned that they study “polyQ-containing proteins that have evidence to be intrinsically disordered (ID) hub-proteins that found in both nucleus and cytoplasm. Do authors find intrinsically disordered regions? If this is already reported, then authors should mention and explain important IDPs in the manuscript. Authors should also explain the importance of IDPs in the nucleus- and cytoplasm-localized hub proteins.

-       Authors search in the database HIPPIE for the PPIs, why did not use other resources such as STRING or MyProteinNet? Authors should use other databases and compare all the interactions.

-  Since IDPs plays important role in PPI, Authors can also use other databases such as D2P2 that show predicted protein binding sites associated with disordered regions (MoRFs) and post-translational modification (PTMs) sites and the location of conserved functional domains. This will give more insights into the PPI study.

Round 2

Reviewer 1 Report

The revised manuscript is well-written, and the authors have addressed my previous comments in this revised version. I recommend publishing this manuscript after the following minor revisions.

Line 22: ubiquitination (NOT ubiqitination)

Line 253: follow (NOT follows)

Reviewer 2 Report

I think that the authors have made sufficient changes to the revised manuscript to improve its quality. It can be accepted for publication.